# Phytochemicals from Purwoceng (*Pimpinella pruatjan*) and Their Potential in Chronic Disease Prevention: Focus on Kidney Health

**DOI:** 10.3390/ijms26178404

**Published:** 2025-08-29

**Authors:** Trisni U. Dewi, Kartika W. Rukmi, Fachrur R. Mahendra, Fauziyatul Munawaroh, Yusuf Ryadi, Leliana N. Widi, Naufal M. Nurdin, Irmanida Batubara

**Affiliations:** 1Faculty of Medicine, IPB University, Bogor 16680, Indonesia; trisniuntari@apps.ipb.ac.id (T.U.D.); kartikarukmi@apps.ipb.ac.id (K.W.R.); munaf@med.kobe-u.ac.jp (F.M.); yusufryadi27@apps.ipb.ac.id (Y.R.); naufal@apps.ipb.ac.id (N.M.N.); 2RSUD Cibinong Hospital, Cibinong 16914, Indonesia; 3Department of Biochemistry, Faculty of Mathematics and Natural Sciences, IPB University, Bogor 16680, Indonesia; rizalmahendra@apps.ipb.ac.id; 4Tropical Biopharmaca Research Center, IPB University, Bogor 16128, Indonesia; lelianawidi22@apps.ipb.ac.id; 5Department of Chemistry, Faculty of Mathematics and Natural Sciences, IPB University, Bogor 16680, Indonesia

**Keywords:** antioxidant, creatinine level, kidney weight, nephrotoxicity, *Pimpinella pruatjan*, urea level

## Abstract

This study examines the potential of purwoceng (*Pimpinella pruatjan*) to prevent chronic kidney damage through its antioxidant and anti-inflammatory properties. To evaluate both efficacy and safety, purwoceng extract was assessed for its phytochemical content and administered to five groups of rats: a healthy control group, a cisplatin-induced nephrotoxicity model, and three treatment groups receiving purwoceng at doses of 20, 30, and 40 mg/kg BW/day. In silico predictions were used for confirmation of in vitro and in vivo results. Renal function was monitored through serum creatinine and urea levels before and after treatment, while kidney tissue was evaluated histopathologically. The results indicate that purwoceng extract complies with safety standards. Notably, a dose of 20 mg/kg BW/day improved glomerular structure in cisplatin-exposed rats, suggesting a nephroprotective effect possibly mediated by vasodilatory and antioxidant mechanisms. In contrast, higher doses (30–40 mg/kg BW) increased urea and creatinine levels, and histological signs indicated only tubule damage. In silico predictions indicate that pinellic acid strongly binds to Cyclooxygenase-2 and Inducible Nitric Oxide Synthase, suggesting its anti-inflammatory potential and benefits for glomerular structure. Therefore, the bioactive compounds in purwoceng may help prevent chronic kidney disease, emphasizing the need for careful dose regulation to avoid toxicity.

## 1. Introduction

Purwoceng (*Pimpinella pruatjan*) is a plant native to Indonesia whose natural habitat is in the Dieng Plateau, Central Java, Indonesia. This plant belongs to the genus *Pimpinella*, which is a cosmopolitan genus and is widespread in various parts of the world. The genus *Pimpinella* is believed to have originated in Eurasia, specifically the regions of Southern Europe, West Asia, to Central Asia. Some of its popular species include *Pimpinella anisum*, which comes from the Eastern Mediterranean (Egypt, Turkey, Syria). *Pimpinella major*, *P. saxifraga*, and others are widely found in Eastern Europe and the Caucasus. *Pimpinella* has undergone local adaptation and speciation, resulting in the emergence of endemic species, such as purwoceng (*Pimpinella pruatjan*), often referred to as the “Viagra of Asia.” Reports suggest that purwoceng enhances both ejaculation and penetration due to its vasodilatory effects, which increase the expression levels of endothelial nitric oxide synthase (eNOS) [1,2,3,4]. The rising interest in its bioactive constituents highlights its advantages for dilating peripheral blood vessels, which can be beneficial for kidney health [5,6]. Our study focuses specifically on kidney health, where oxidative stress and inflammation are central drivers of injury. In the kidneys, xenobiotic exposure (e.g., cisplatin) can co-activate inducible nitric oxide synthase (iNOS) and Cyclooxygenase-2 (COX2), while NADPH Oxidase-5 (NOX5) amplifies reactive oxygen species (ROS), collectively sustaining inflammatory and oxidative damage [1].

Purwoceng contains antioxidants that can reduce oxidative damage, which is a known cause of inflammation, metabolic problems, damaged blood vessels, and impaired nerve function [2]. However, its contents can also interfere with liver and kidney function at inappropriate doses. Therefore, it is important to determine the optimum safe dose for kidney function and determine the compound that causes damage to kidney function [3].

We selected the cisplatin model because its injury drivers, oxidative stress and inflammatory signaling via NOX-derived ROS and the COX2/iNOS axis, closely match the predicted targets of purwoceng metabolites identified by LC-MS and in silico docking (notably pinellic acid). This mechanistic alignment, together with the model’s reproducibility and clinical relevance, supports its use to test our hypothesis. One compound that can damage the kidneys is cisplatin. Cisplatin has been used extensively in the treatment of a variety of solid tumors, including breast, lung, head, and neck cancer; however, because of its harmful side effects on the liver and kidneys, its use has been restricted [4,5]. Cisplatin can damage cell membranes and induce tubule damage through the mechanism of formation of ROS (reactive oxygen species) and hydroxyl radicals, which can cause lipid peroxidation, inflammation, and hypoxia [5,6,7]. The damage process that occurs can cause a reduction in the glomerular filtration rate and cause acute toxicity in the kidneys [5]. Cisplatin accumulates in the kidneys and extends the half-life of cisplatin to 58–73 min from 25–49 min [8]. The efficacy and side effects that occur can depend on the dose and duration of use. The administration of low doses of cisplatin causes apoptosis, while high doses of cisplatin cause necrosis [9,10].

An in silico approach reveals that xenobiotic compounds, such as drugs, pesticides, and other foreign chemical compounds, can trigger kidney inflammation by activating the nitro-oxidative pathway. Chronic exposure to xenobiotics stimulates the expression of inducible nitric oxide synthase (iNOS) and cyclooxygenase-2 (COX2), two pro-inflammatory enzymes that work synergistically to strengthen the immune response. The interaction between the pro-inflammatory enzymes COX2 and iNOS plays an important role in amplifying the inflammatory response. iNOS is known to bind directly to COX2 and cause S-nitrosylation, which increases COX2 activity and the production of inflammatory mediators, such as prostaglandin E2 (PGE2) [11]. Disruption of this interaction has the potential to suppress COX2 activation mediated by nitric oxide (NO), making it a promising therapeutic target for the development of anti-inflammatory drugs. In another pathway, NOX5 (NADPH Oxidase-5) acts as the primary producer of reactive oxygen species (ROS), which exacerbate inflammation through the activation of signaling pathways, such as NF-κB, which in turn stimulates the expression of COX2 and matrix metalloproteinases [12]. Chronic activation of NOX5, iNOS, and COX2 has been closely associated with the development of chronic inflammatory diseases and cancer [13]. Furthermore, the complex interaction between ROS and RNS (reactive nitrogen species) in inflammatory conditions leads to nitro-oxidative stress, which can damage tissues and exacerbate chronic inflammatory conditions [14]. In this context, antioxidant compounds are believed to neutralize oxidative stress, restore control of immune function, and have the potential to be developed as therapeutic agents in the management of inflammation-related diseases [15].

The efficacy of the extract is related to the phytochemical compounds in the extract. The consistency and quality of the extract can be determined by its total phenolic content, total flavonoid content, and specific compounds, which can be determined using chromatography techniques. Therefore, this research aims to determine the effect of purwoceng ethanolic extract, as well as its phytochemicals, on cisplatin-induced rats by assessing their kidney function based on blood creatinine levels, blood urea nitrogen (BUN), kidney organ weight, clinical features, and histopathological examination. In addition, in silico prediction tests were used to confirm the results of in vivo testing, exploring putative mechanisms by docking identified constituents against iNOS, COX2, and NOX5. This mechanistic alignment focuses this paper on kidney protection rather than ethnomedical claims of sexual function.

## 2. Results

### 2.1. Purwoceng Root Extract

Purwoceng root extract was produced using 96% food-grade ethanol as the solvent, with an average yield of 5.29%. The extract obtained was a paste with a greenish-brown color and a distinctive purwoceng odor. The extract’s safety is determined against heavy metal content and contamination by pathogenic bacteria. The results are summarized in Table 1 for heavy metal and Table 2 for pathogenic bacteria. All safety parameters required by the Indonesian Food and Drug Supervisory Agency (BPOM) and WHO have met the standards, with levels lower than the detection limit of the AAS tool and no presence of pathogenic microbes [16,17].

The phytochemical contents of purwoceng ethanol extract were determined based on its characteristic compounds, including total phenolic and flavonoid contents, as well as individual phytochemicals identified through LC-HRMS analysis. The results are summarized in Table 3, with the raw LC-HRMS data provided in the Appendix A. Approximately 100 peaks were detected in the LC-HRMS analysis, of which 23 were identified as major peaks. Eight of these major peaks could not be annotated and are therefore classified as unknown compounds. The remaining 15 annotated compounds belong to the carbohydrate, phenolic, and fatty acid groups. Among them, juniperic acid was the most abundant compound in the extract.

The ability of purwoceng ethanol extract to destroy DPPH radicals and ABTS can be seen in Table 4. The results show that the extract can react to DPPH radicals, but it is not better than vitamin C as a positive control. Likewise, with the ability to react with ABTS radicals, the ability of vitamin C is 34 times better than that of purwoceng extract.

### 2.2. Baseline and Post-Treatment Characteristics of the Rats

A total of 35 rats were initially included in the study, but only 30 were analyzed. During the cisplatin induction and purwoceng treatment, five rats died: one in the P20 group in the cisplatin induction phase, two in the P30 group in the purwoceng treatment phase (days 5–7 of treatment), and two in the P40 group in the purwoceng treatment phase (day 5 of treatment). There were no statistically significant differences in baseline urea and creatinine levels among the five groups (*p* > 0.05) (Figure 1). Deaths occurred within 48–72 h after cisplatin injection and within the purwoceng treatment phase, which corresponds to the reported peak of cisplatin accumulation and nephrotoxicity in the kidney. Clinical signs observed before death included reduced food intake, lethargy, and urinary difficulties, consistent with acute cisplatin toxicity and renal histopathology.

We compared the rats that received only cisplatin before treatment with the rats in the control group. Rats that received cisplatin at a dose of 7 mg/kg BW had elevated creatinine and urea levels. Urea levels were significantly elevated in the cisplatin group compared with the control group, while creatinine levels showed no significant differences. Average urea levels were higher in the group treated with purwoceng extract at doses of 20 mg/Kg BW (P20), 30 mg/Kg BW (P30), and 40 mg/Kg BW (P40) compared to the control group, with P20 showing no significant difference from the cisplatin group. However, P30 and P40 had significantly elevated urea compared to cisplatin. The P40 group also had significantly higher creatinine levels compared to the control group (Figure 2A,B). In terms of kidney weight, the cisplatin and P20 groups were significantly heavier than the controls, while the P30 and P40 groups showed no significant differences in weight compared to the controls (Figure 2C,D). While rats in the cisplatin group had similar clinical characteristics to those in the control group, rats in the P30 and P40 groups exhibited fatigue, epistaxis, and prolapsed penises.

### 2.3. Histopathological Renal Appearance

Rats in the control rat group had several synechiae and showed vasoconstriction of not more than 10% of the total number of glomeruli; thus, kidney function was considered to still be within control limits and was therefore assigned an injury score of zero (Figure 3a). Likewise, for the tubules, there was no epithelial thinning, intraluminal cast, tubule dilatation, or inflammation, and the brush border was present on more than 95% of the tubule lumen surface; thus, the injury score was zero. Rats that received cisplatin showed a histopathological picture of renal damage in the form of glomeruli with synechiae to Bowman’s capsule in fewer than half of the glomeruli, and some of the capillaries appeared closed; thus, they were assigned a glomerular injury score of 2. Renal tubules in rats administered cisplatin showed inflammation, and intraluminal casts were found in less than 25% of the entire visual field, while brush borders were found in almost 75% of the entire tubule lumen surface; thus, they were assigned an injury score of 1 (Figure 3b). Rats treated with various doses of purwoceng (20, 30, and 40 mg/kg BW) after cisplatin-induced nephrotoxicity showed similar histopathological pictures, i.e., improved glomerular injury scores (Figure 3c–e). Where previously the glomerulus in cisplatin-treated kidneys showed that more than half of the glomerular capillaries were closed, after treatment with purwoceng, almost 100% of the capillaries in the glomerulus were open, and only glomerular synechiae to the Bowman’s capsule were still present. Rats with cisplatin-induced nephrotoxicity in the P20 group had lower tubular injury scores compared with rats in the P30 and P40 groups (Figure 3d,e).

### 2.4. In Silico Prediction Target Radical Enzyme

#### 2.4.1. Structure Align

In silico testing helps to identify which bioactive compounds cause in vivo phenomena to occur, based on the active compounds identified by LC-MS (Table 3). The results of the ligand affinity ranking against three enzyme targets show interesting variations in interaction potential. In this case, the study employed an in silico approach focusing on enzymes involved in free radical production, namely cyclooxygenase−2 (COX2) and nitric oxide synthase (iNOS) from *Homo sapiens*, as well as NADPH oxidase (NOX5) from *Cylindrospermum stagnale*. Enzymes from different organisms were employed relative to *Mus musculus* in order to bridge therapeutic modeling from mouse to human and to utilize native ligands within the protein structures as comparative references. The BLASTp results revealed that orthologous enzymes from *Mus musculus* exhibited a high degree of identity with human iNOS and COX2, whereas NOX5 from *C. stagnale* showed relatively low similarity (Table 5). On the other hand, 3D structural superposition between *Mus musculus* enzymes and their orthologs demonstrated nearly identical conformations, as reflected by low root mean square deviation (RMSD) values, where an RMSD close to 2 Å indicates significant structural similarity (Table 6, Figure 4).

#### 2.4.2. Docking Validation

The docking procedure was initiated by validating the grid box parameters. In this study, redocking was performed 50 times, ranging from the smallest grid box size of 10 × 10 × 10 Å to 50 × 50 × 50 Å, with a spacing of 1 Å, using the native ligands of each enzyme. For COX2, redocking was conducted with the native ligand salicylic acid at the coordinates x = 41.56, y = 24.16, z = 240.29. The results show an average binding energy of −5.87 ± 0.04 kcal·mol^−1^ and an average RMSD of 0.34 ± 0.007 Å. The optimal grid box, with the lowest RMSD value (0.34 Å), was obtained at a size of 49 × 49 × 49 (Å). For iNOS, redocking was performed using the native ligand ethyl 4-[(4-methylpyridin-2-yl)amino]piperidine-1-carboxylate at the coordinates x = 37.19, y = 34.78, z = 32.03. The analysis yielded an average binding energy of −9.60 ± 1.24 kcal·mol^−1^ and an average RMSD of 2.19 ± 0.34 Å. The best grid box (lowest RMSD: 0.90 Å) was observed at a size of 24 × 24 × 24 (Å). For NOX5, redocking was carried out with the native ligand 8-[2-(4-cyclohexylphenyl)quinolin-4-yl]carbonyl-1,3,8-triazaspiro[4.5]decane-2,4-dione at the coordinates x = 36.70, y = −55.04, z = 31.08 Å. The results indicate an average binding energy of −11.80 ± 0.66 kcal·mol^−1^ and an average RMSD of 1.49 ± 0.75 Å. The optimal grid box (lowest RMSD: 0.59 Å) was obtained at a size of 45 × 45 × 45 (Å).

#### 2.4.3. Molecular Docking

Against COX2, pinellic acid showed the highest affinity with a value of −11.23 kcal/mol, significantly outperforming the reference compound salicylic acid (−5.88 kcal/mol) (Figure 5a). Additionally, conjugated unsaturated fatty acids, such as 9Z,11E,13-Oxo-octadecadienoic acid and 10E,12Z,9-Hydroperoxy-octadecadienoic acid, also exhibited strong interaction potential (≤−10 kcal/mol). For the inducible nitric oxide synthase (iNOS) target, pinellic acid again ranked first with an affinity of −9.09 kcal/mol, followed by sebacic acid (−8.82 kcal/mol) (Figure 5b). The reference drug, Ethyl 4-[(4-methylpyridin-2-yl)amino]piperidine-1-carboxylate, has an affinity of −8.74 kcal/mol, placing it below the two main compounds. This suggests that some natural compounds in this study may be more effective than synthetic drug compounds in inhibiting iNOS. Meanwhile, for NOX5, setanaxib, as the reference drug, showed the highest affinity of −9.72 kcal/mol, closely followed by 10E,12Z,9-Hydroperoxy-octadecadienoic acid (−9.31 kcal/mol). The compounds 13S-hydroxyoctadecadienoic acid and 9Z,11E,13-Oxo-octadecadienoic acid also showed strong interactions, with affinities of −8.34 and −8.15 kcal/mol, respectively. In contrast, compounds such as Hexitol (−5.16 kcal/mol) and pinellic acid (−6.20 kcal/mol) exhibited weaker affinities toward this target (Figure 5c). Overall, pinellic acid and several oxidized unsaturated fatty acid derivatives emerged as potential multi-target candidates, showing high affinity for two of the three enzyme targets.

In the COX2 enzyme, salicylic acid (Figure 6a) forms two main hydrogen bonds with the Gly526A and Trp387A residues. These interactions are simple and characteristic of small aromatic acid-based compounds. In contrast, pinellic acid (Figure 6d) exhibits more interactions with active site residues, including Val523A, Ser353A, and Arg120A. There are extensive hydrogen interactions and ionic interactions with the guanidino group of Arg120A. Additionally, hydrophobic contacts with Leu352A and Tyr385A enhance ligand affinity. The complexity and diversity of these interaction types suggest that pinellic acid could be a potential COX2 inhibitor with a multisite binding mechanism. For the iNOS enzyme, the reference ligand (Figure 6b) shows significant ionic interactions with Glu377D, two π-π stacking interactions with aromatic groups, and several hydrogen bonds with Pro350D and Val352D. In contrast, pinellic acid (Figure 6e) again shows strong hydrogen-based interactions with residues Tyr373D, Gln263D, and Tyr347D. Although there are no aromatic interactions, the test ligand stabilizes binding through three negatively charged hydroxyl and carboxylate groups, indicating potential inhibition via a polar binding mechanism.

In the NOX5 enzyme, setanaxib (Figure 5c) forms strong hydrogen interactions, π-π interactions, and hydrophobic contacts with residues Tyr347D, Cys668A, and Ile538A, indicating a combination of aromatic and polar interactions. In contrast, the test ligand 10E,12Z,9-hydroperoxy-10,12-octadecadienoic acid (Figure 6f) interacts extensively with residues Gly669A, Pro542A, and Asn692A through hydrogen bonds and hydrophobic contacts from its hydrocarbon chain. Although it lacks aromatic groups, this ligand exhibits structural flexibility that allows it to explore the enzyme’s active site through comprehensive polar and hydrophobic interactions.

## 3. Discussion

The safety of purwoceng extract (Table 1 and Table 2) follows previous research reports, which show that it meets standards and is safe to use as an ingredient for in vivo tests [18]. This means that the purwoceng root used in this study was grown in high-quality soil and water, and underwent adequate processing, transportation, and storage [19]. In addition to safety, the phytochemical constituents found in the ethanolic extract of purwoceng, such as total phenolic and flavonoid contents (see Table 3), were consistent with the ranges reported in previous studies: 0.55% to 2.95% gallic acid equivalent per purwoceng extract and 0.34% to 0.98% quercetin equivalent per extract [6]. Furthermore, in vivo testing demonstrated improvements in kidney function in the P20 group. Histopathological results also indicate enhancements in the glomerular score, suggesting a potential anti-inflammatory effect. The in silico confirmation results reveal that the extract contains phenolic compounds such as ferulic acid, pinellic acid, and 3,4-dihydroxyphenylacetic acid, as well as unsaturated fatty acids, including (10E,12Z)-9-Hydroperoxy-10,12-octadecadienoic acid, 13S-hydroxyoctadecadienoic acid, linoleic acid, oleic acid, and ricinoleic acid (see Table 3 and Appendix A). These compounds have a strong binding affinity with COX2 and eNOS, providing anti-inflammatory effects. The phenolic compounds and unsaturated fatty acids are responsible as antioxidants [20], although these levels were lower than those of vitamin C in this study. Other compounds that were detected include juniperic acid, which has been detected as an abundant component (Table 3), but studies on its biological activity, including its anti-inflammatory effects that can improve kidney function, are very limited [21,22]. Other studies have shown that juniperic acid has biological activity as an antibacterial [23].

While several *Pimpinella* species exhibit medicinal properties, *P. pruatjan* demonstrates unique advantages for nephroprotection. Compared to the well-studied *P. anisum* (rich in anethole but lacking renal-specific effects) [24], our LC-MS analysis revealed *P. pruatjan*’s distinctive composition of pinellic acid and juniperic acid (Table 3), which showed strong binding to renal inflammatory targets COX2 and iNOS (Figure 5). Notably, at just 20 mg/kg BW, *P. pruatjan* improved glomerular histopathology (Figure 3c). These findings position *P. pruatjan* as a uniquely balanced nephroprotective agent, combining the anti-inflammatory specificity of synthetic drugs with the safety profile of herbal medicines.

In this study, kidney function examinations were carried out, which included monitoring indicators, such as serum urea and creatinine concentrations. A reduction in glomerular filtration rate (GFR) typically leads to elevated levels of these markers in the bloodstream, which occurs only after significant (roughly 30%) kidney damage [25]. In this study, 7 mg/kg BW of cisplatin increased urea and creatinine levels and kidney weight after 3 days, indicating renal failure, consistent with previous findings [26]. Cisplatin concentration levels reach peaks in rats 48–72 h after administration, with significant urinary elimination within the first 72 h. Its half-life changes over time [26,27]. Inflammation is a key contributor to cisplatin-induced acute kidney injury [26]. Inflammation and edema likely cause increased kidney weight, as shown in this study. In addition, histopathological examination showed damage to the glomerulus (glomerular injury score 2) and inflammation of the tubules (tubular injury score 1) (Figure 3c). Cisplatin (3–8 mg/kg BW) causes minimal changes within 1–2 days, with more significant damage, such as brush border loss and cell necrosis, appearing after 3–4 days [25]. Cisplatin concentrates mainly in the mitochondria at 72 h. The renal sample was assigned a code by a veterinarian and examined by a histopathologist. The code was revealed after the histopathologist completed the examination. Clinically, rats initially showed no signs of cisplatin administration, possibly because the extent of kidney damage depends on the dosage, frequency, and duration of exposure. A single 7.5 mg/kg BW cisplatin dose causes less reduction in kidney function than repeated dosing of the same amount, and clinical signs may not be immediately apparent. The lethal dose in rats is 16 mg/kg BW [27].

The purwoceng extract dose of 20 mg/kg BW in this study (P20) showed lower urea and creatinine levels compared with the cisplatin groups, which only received cisplatin for nephrotoxicity induction (Figure 2A,B). The microscopic findings in the P20 groups showed an improvement in the glomerular score, which became 0, while in the cisplatin group, it remained 2. The tubular score was similar to that of the negative control (Figure 3b,c). This can be explained by the phenolic and its unsaturated fatty acid compounds, such as pinellic acid, which were detected putatively through LC-HRMS analysis (Appendix A). Coupled with the in silico results, this indicates that pinellic acid had the strongest affinity for cyclooxygenase-2 and inducible nitric oxide synthase (Figure 4). Phenolic compounds are reported to be responsible for the antioxidant [19]. Purwoceng root extract demonstrated low antioxidant activity based on its radical scavenging against DPPH and ABTS (Table 4), but these compounds have a role in counteracting free radicals through either direct mechanisms, by reducing ROS and nitrogen radical species (RNS), or indirect mechanisms, by inhibiting pro-oxidant enzymes and activating antioxidant enzymes [28,29,30]. The compounds in the extract that are possible antioxidants are the unsaturated fatty acids (Table 3). Unsaturated fatty acids, such as ricinoleic, 10E,12Z-9-Hydroperoxy-10,12-octadecadienoic acid, and oleic acid, which were detected by LC-MS (Appendix A), have benefits as non-polar antioxidants that can counteract hydrophobic radical species, such as lipid peroxyl radicals, and there is evidence that unsaturated fatty acids have benefits as anti-inflammatory tissue in kidney cells, thereby reducing the risk of CKD [31,32,33]. These results indicate that the antioxidant activity in purwoceng provided a defense mechanism against cisplatin-induced cellular kidney damage caused by reactive oxygen species formation in the negative control group. Once cisplatin entered these cells, it induced the formation of reactive oxygen species that cause cellular damage, including DNA damage, inflammation, and eventually cell death. In addition, the active compounds in purwocheng root extract demonstrated the ability to effectively interact with three primary inflammatory targets. The diverse types of bonds displayed in the visualization reinforce the hypothesis that the test ligand has pharmacological potential comparable to or even higher than the reference ligand, depending on the enzymatic context and in vivo bioactivity [34].

On the other hand, based on a previous study, flavonoids and alkaloids found in purwoceng root extract are lipophilic compounds [35]. In this study, these compounds were not identified, but may be part of an unknown group shown in the Appendix A, as detected by LC-HRMS analysis of purwoceng ethanolic root extract. The lipophilic compounds require enzymatic conversion into hydrophilic forms before they can be excreted in urine. The more lipophilic a compound is, the more challenging it becomes for the kidneys to excrete it [35]. These compounds are challenging to eliminate through the kidneys; therefore, if administered for an extended period and at higher doses, they will build up in the proximal tubule. The proximal tubule exhibits a high sensitivity to toxic substances. This may explain why higher doses of purwoceng (P30 and P40) increased the urea and creatinine levels, indicating a nephrotoxic effect, which is also reflected by the tubule score. This finding needs a dose adjustment. Clinically, rats receiving cisplatin and higher doses of purwoceng (30–40 mg/kg BW/day) showed decreased activity, weakness, pain, nasal and eye bleeding, penile prolapse, urination difficulties, and bladder sediment. These conditions may be related to the effects of cisplatin induction, which increases ROS and is further enhanced by the combination of lipophilic components in purwoceng, which further damages the kidneys, especially the tubules.

A note of this research is that this study was conducted over a relatively short period (7 days), and there are unknown components that were not detected that could potentially influence the results. We also did not perform pharmacokinetic/tissue distribution studies, quantitative tissue compound measurements, or a formal repeated-dose toxicology battery, nor did we measure renal injury biomarkers (e.g., KIM-1, NGAL) or tissue oxidative-stress markers (MDA, SOD, GPx). These experiments are required to confirm whether higher doses cause direct nephrotoxicity, alter cisplatin pharmacodynamics, or produce non-linear exposure/toxicity relationships, and are planned for follow-up work.

## 4. Materials and Methods

### 4.1. Study Design and Setting

This experimental study used a control group design with pre- and post-test analysis to measure creatinine and urea levels in serum, and a post-test-only control group design to assess renal weight and renal histopathology. The study was conducted between July and October 2024.

### 4.2. Sample Preparation and Extraction

Purwoceng root was collected from Banjarnegara, Central Java, Indonesia. The plant was identified previously [36]. The root was dried in an oven at 50 °C, and the powder was extracted with food-grade ethanol (ratio of a 1 g sample to 10 mL ethanol) by maceration for 24 h at room temperature. The extract was then concentrated using a rotary evaporator at a temperature of 40 °C, protected from light, and not stirred, but shaken 3 times over 24 h. The extract obtained was standardized for its safety and compounds. The microbial safety was determined based on the WHO method No. 18, including the total plate count, yeast, and coliform. The arsenic and heavy metal content were determined using the WHO method No. 17 [17]. Roots of purwoceng were extracted using ethanol at a ratio of a 1 g sample to 10 mL ethanol (0.1 g/mL) [13]. The resulting extracts were then pooled and used for analysis of the total phenolic content, total flavonoids, and antioxidant capacity.

### 4.3. Phytochemical Content

Phytochemical analysis was carried out based on Batubara et al. [37]. For the total phenolic content, 20 μL of the sample or standard gallic acid was reacted with 120 μL of 10% Folin–Ciocalteu reagent, and the mixture was incubated for 5 min. Following this, 80 μL of Na_2_CO_3_ 10% was added, and the mixture was incubated for 30 min using nano spectrophotometry (SPECTROstar Nano BMG LABTECH), and absorbance was measured at 750 nm. The gallic acid equivalent (GAE) concentration parameters were used to determine the total phenolic content. Then, for the total flavonoid content, 120 μL of distilled water was mixed with 10 μL of sample or quercetin standard, 10 μL of 10% AlCl_3_, 10 μL of glacial acetic acid, and 50 μL of ethanol that had been pre-analyzed. The mixture was then incubated for 30 min. The absorbance was measured with nanospectrophotometry (SPECTROstar Nano BMG LABTECH, Ortenberg, Germany) at a wavelength of 415 nm. The quercetin equivalent (QE) concentration characteristics were used to calculate the total flavonoid content.

### 4.4. Antioxidant Capacity

Antioxidant activity was determined using the 1,1-diphenyl-2-picrylhydrazyl (DPPH) and 2,2′-azino-bis (3-ethylbenzothiazoline-6-sulfonic acid) (ABTS) techniques based on the methods of Prayogo et al. [38]. In a 96-well plate, 100 µL of the extract solution was mixed with 100 µL of DPPH radical solution in methanol at a concentration of 125 µM. The mixture was incubated for 30 min at room temperature (27–28 °C). Absorbance was then measured at 515 nm using a microplate reader (Epoch Biotek, Santa Clara, CA, USA). A Trolox calibration curve served as the positive control, and the antioxidant activity was expressed as mmol Trolox equivalents per gram of extract (mmol TE/g Ext). For the ABTS assay, 5 mL of 7 mM ABTS reagent was combined with 88 µL of 140 mM K_2_S_2_O_8_ and left in a dark room for 16 h. The solution was then diluted with a mixture-to-water ratio of 1:44 (*v*/*v*). In a 96-well plate, 180 µL of the ABTS reagent was mixed with 20 µL of extract solution and incubated for 6 min at room temperature. After incubation, absorbance was recorded at 734 nm using a microplate reader (Epoch Biotek). The antioxidant capacity, representing scavenging activity, was calculated using a trolox calibration curve and reported as mmol TE/g Ext.

### 4.5. Compound Identification in Purwoceng Root Extract

The ethanolic extract (2 µL) was injected using the UHPLC Vanquish Tandem Q Exactive Plus Orbitrap HRMS ThermoScientific. The column used was Accucore C18, 100 × 2.1 mm, 1.5 μm (ThermoScientific, Waltham, MA, USA) at 30 °C. The mobile phase used was (A) 0.1% formic acid in water and (B) acetonitrile with 0.1% formic acid, at a flow rate of 0.2 mL/min. The gradient program used was 0–1 min (5% B), 1–25 min (5–95% B), 25–28 min (95% B), and 28–33 min (5% B). The column temperature was 30 °C. The mass range collected was from 100–1500 *m*/*z* in negative ion mode at a resolution of 17,500. The MS spray voltage was 3.80 kV, with a capillary temperature of 320 °C. The data obtained was processed using online databases from ChemSpider and mzCloud [39,40]. The quantitative analysis of the putative compounds was determined based on the % area of the related peak.

### 4.6. In Silico Prediction

AutoDock-GPU v1.6, based on [41], was also used to validate the docking of 15 ligands from LC-MS/MS to several target enzymes related to free radical mechanisms, such as NADH oxide (8CAO), inducible nitric oxide (3E7G), and cyclooxygenase-2 (5F1A) (https://doi.org/10.1021/acs.jctc.0c01006). The receptor and the 15 ligands were first converted into PDBQT format using the ADFRsuite v1.0 program (https://doi.org/10.1371/journal.pcbi.1004586). Docking with the AutoDock-GPU program was performed within a Google Colab Notebook. The grid box was set to the center of each co-crystal ligand in PDB structures, with dimensions ranging from 10 × 10 × 10 to 60 × 60 × 60 Å. The grid parameter file (GPF) for each ligand was generated using MGLTools v1.5.7 and executed with AutoGrid4. AutoDock-GPU implements the Lamarckian genetic algorithm (LGA) by default. The number of LGA runs and the population size were both set to 100. The best ligand position, with its corresponding binding affinity, was generated as the output. The RMSD of the docked ligand position relative to the initial position was calculated using RDKit v2025.3.3. Visualization of protein–ligand complexes in 2D was carried out using the Protein Plus webserver (https://proteins.plus/, accessed on 24 August 2025).

### 4.7. Inclusion and Exclusion Criteria

Male Sprague Dawley white rats were used based on the criteria of Arjadi et al. [42]: male, body weight of approximately 150–200 g, aged between 8 and 10 weeks, and healthy with no anatomical or behavioral abnormalities. Rats were excluded if they became ill or died during the study or if their body weight increased or decreased by more than 20% during acclimatization.

### 4.8. Experimental Protocol

Based on Federer’s formula [43], a total of 35 rats were used, split into 5 groups. One group received no treatment at all (Control). One group received cisplatin only (Cis) at a dose of 7 mg/kg BW/day. Three groups received cisplatin and purwoceng root extract at different doses, including purwoceng doses of 20 mg/kg BW/day (P20), 30 mg/kg BW/day (P30), and 40 mg/kg BW/day (P40), with treatment given 3 days after of cisplatin induction. The dose range from 20 to 40 mg was chosen based on a study conducted by Arjadi, where the dose of purwoceng root extract given subchronically (*Pimpinella pruatjan*) induced hepatotoxicity at a dose of 42 mg/Kg BW. The animals were acclimatized to the laboratory environment for seven days before the study began. Throughout the treatments, a fixed dosage of purwoceng root extract was administered orally every day for seven days. During the treatment period, the health and mortality of the animals were monitored daily. In addition, food and water intake and weight were tracked every week. Blood samples were collected from the animals at the beginning and end of the study to evaluate the urea–creatinine levels. Additionally, close examination of the renal tissue samples under a microscope was carried out to determine histological characteristics.

### 4.9. Data Collections

The study data included measurements of the blood urea–creatinine levels, renal weight, clinical features, and histological features of the kidneys based on the methods of Arfian et al. [44]. The urea levels were examined using the urease-GLDH method at λ340 nm, and serum creatinine was examined using the Jaffe method at λ492 nm, which was read on a spectrophotometer. The rat kidney samples for histopathological examination were labeled by a veterinarian, keeping the histopathologist unaware of which rats were treated or untreated. The extent of renal histological damage was assessed using the PAS tubular and glomerular score. Control cells received a score of 0 for *tubule* injury. Scores were based on the extent of damage: 1 for *tubule* injury in <25% of the visual field; 2 for *tubule* injury in 25–50% of the visual field; 3 for *tubule* injury in 51–75% of the visual field; and 4 for *tubule* injury in >75% of the visual field. A glomerular score of zero was assigned to the control, meaning that there were no synechiae and all capillaries were open; a score of 1 indicated capillaries closure of less than 25%, with synechiae present; 2 indicated capillary closure of <50%; 3 indicated capillary closure of <75%; 4 indicated capillaries closure of >75%.

### 4.10. Data Analyses

Shapiro–Wilk’s test was used to determine the control of the data; the Kruskal-Wallis test was not used. Since the data were found to be normally distributed and homogeneous, the parametric-repeated ANOVA method was also applied. Additionally, a post hoc test with the least significant difference (LSD) was conducted to determine the significance of the differences among the treatment groups.

## 5. Conclusions

Purwoceng root extract at a dose of 20 mg/kg BW/day improved glomerular histopathology after cisplatin-induced nephrotoxicity. Pinellic acid, detected in the extract, is suspected to be a bioactive compound that inhibits COX and iNOS, according to in silico analysis. Purwoceng at a dose of 20 mg/kg BW is thought to play a role in improving kidney function due to the administration of the xenobiotic cisplatin, while higher doses are related to nephrotoxicity.

## Figures and Tables

**Figure 1 ijms-26-08404-f001:**
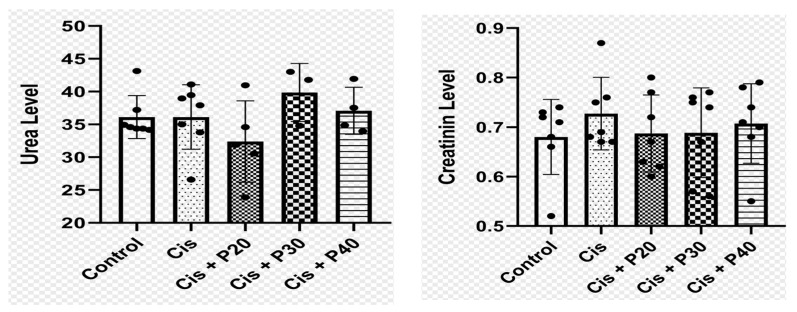
Baseline characteristics of urea and creatinine levels. Each analysis used 5 rats. The results indicate that the groups were not significantly different (*p* > 0.05). Control: no treatment; Cis: cisplatin; P20: 20 mg/kg BW purwoceng; P30: 30 mg/kg BW purwoceng; P40: 40 mg/kg BW purwoceng. Urea and creatinine levels were not different between groups.

**Figure 2 ijms-26-08404-f002:**
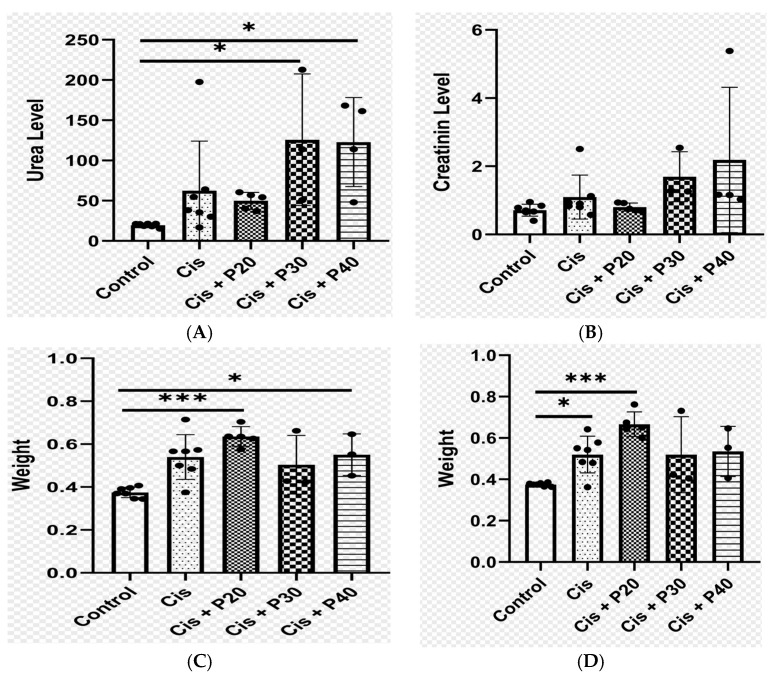
Urea levels (**A**), creatinine levels (**B**), right kidney weights (**C**), and left kidney weights post-treatment (**D**). * indicates significant results (*p* < 0.05); *** indicates significant results (*p* < 0.001). Control: no treatment; Cis: cisplatin; P20: 20 mg/kg BW purwoceng; P30: 30 mg/kg BW purwoceng; P40: 40 mg/kg BW purwoceng.

**Figure 3 ijms-26-08404-f003:**
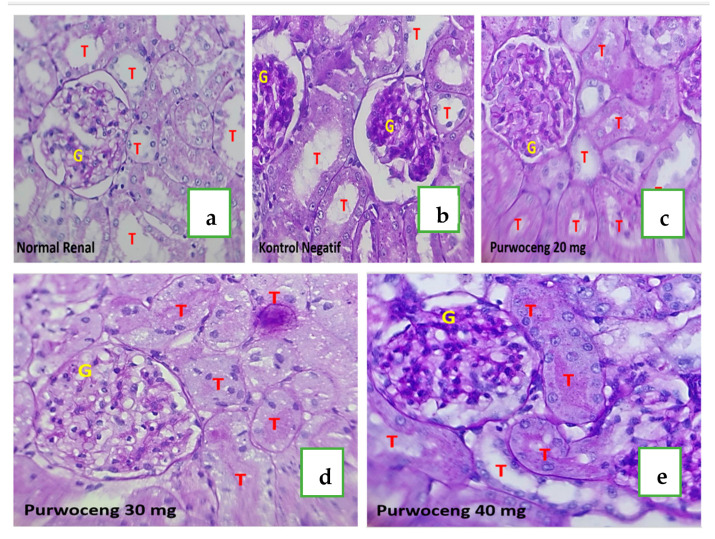
Histopathological images of rat kidneys, glomerulus, and tubules appearance (400× magnification). (**a**) Control rat group, (**b**) cisplatin group, (**c**) P20 group, (**d**) P30 group, (**e**) P40 group. T (red color): tubules, G (yellow color): glomerulus.

**Figure 4 ijms-26-08404-f004:**
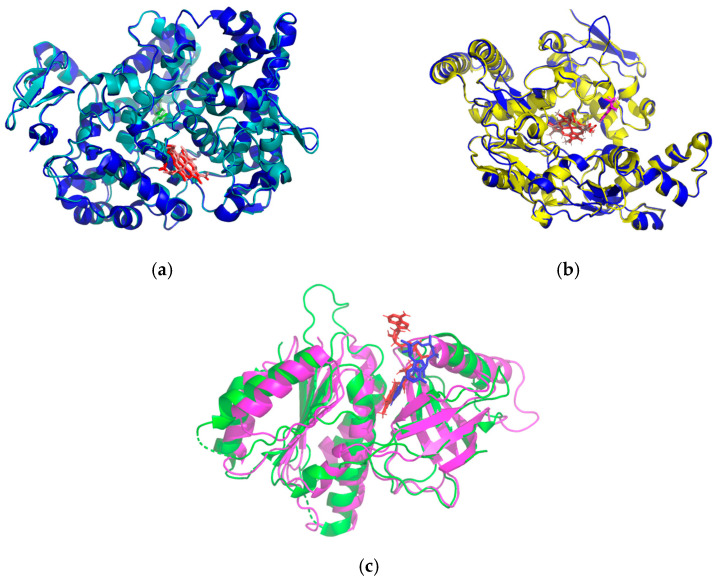
Superpose (**a**) COX2 (*Mus musculus*: cyan, *Homo sapiens*: dark blue), (**b**) iNOS (*Mus musculus*: yellow, *Homo sapiens*: dark blue), (**c**) NOX5 (*Mus musculus*: green, *Cylindrospermum stagnale*: magenta).

**Figure 5 ijms-26-08404-f005:**
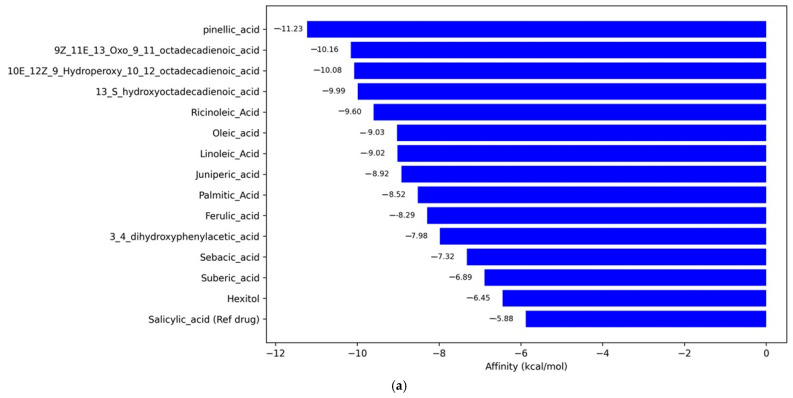
Virtual screening analysis of three types of free radicalpromoting enzymes with purwoceng plant metabolite compounds. (**a**) Cyclooxygenase−2, (**b**) Inducible Nitric Oxide, and (**c**) NADPH oxide.

**Figure 6 ijms-26-08404-f006:**
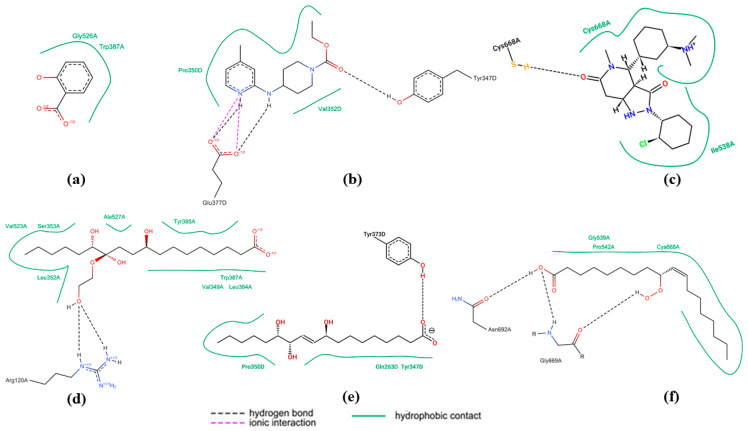
The best 2D interactions based on binding affinity toward three types of enzymes related to free radical production. Reference ligand interactions toward three enzymes: (**a**) salicylic acid-COX2, (**b**) ethyl 4-[(4-methylpyridin-2-yl)amino]piperidine-1-carboxylate-iNOS, and (**c**) setanaxib-NOX5 (in sequence). The best test ligands from the purwoceng plant against three enzymes: (**d**) pinellic acid-COX2, (**e**) pinellic acid-Inos, and (**f**) 10E,12Z,9-Hydroperoxy-10,12-octadecadienoic acid-NOX5. Colored residues indicate amino acids forming hydrogen bonds or hydrophobic contacts with the ligand, while arrows depict stabilizing interactions. Pinellic acid interacts strongly with Tyr385 and Arg120 in COX2 and with Glu371 and Tyr485 in iNOS, consistent with high binding affinity. These interactions suggest suppression of pro-inflammatory pathways.

**Table 1 ijms-26-08404-t001:** Heavy metal safety of purwoceng extract.

Safety Parameter	Standards Permitted by						Result
	BPOM No. 13 of 2019	WHO 2007					
		Malaysia	Canada	Singapore	Thailand	Recommendation	
Heavy metal content safety of purwoceng							
Cd Maximum (mg/Kg)	0.3	-	0.3	-	0.3	0.3	<detection limit (LOD 0.03 mg/kg)
Pb Maximum (mg/Kg)	10	10	10	20	10	10	<detection limit (LOD 0.033 mg/kg)
As Maximum (mg/Kg)	5	5	5	5	4	-	<detection limit (LOD 0.008 mg/kg)

Note: - indicates no information.

**Table 2 ijms-26-08404-t002:** **Pathogenic** microbial content of purwoceng extract.

Safety Parameter	Standards Permitted by		Result
	BPOM No. 13 of 2019	WHO 2007	
		Recommendation	
Total plate count (TPC) maximum (CFU/g)	5 × 10^7^	10^7^	Negative
Coliform maximum (CFU/g)	10^3^	10^2^	Negative
mold/yeast maximum (CFU/g)	5 × 10^5^	10^4^	Negative

**Table 3 ijms-26-08404-t003:** Contents of compounds characteristic of purwoceng ethanol extract.

Compound	Content	Method of Determination
Total phenolic content (% gallic acid equivalent/extract)	2.73	Semi-qualitative using a spectrophotometer UV-Vis
Total flavonoid content (% quercetin equivalent/extract)	0.75	Semi-qualitative using a spectrophotometer UV-Vis
(E)-Ferulic acid (%)	1.26	LC-HR MS
3,4-dihydroxyphenylacetic acid (%)	4.70	LC-HR MS
(−)-pinellic acid (%)	3.32	LC-HR MS
13S-hydroxyoctadecadienoic acid (%)	2.97	LC HR MS
Juniperic acid (%)	7.28	LC HR MS
Linoleic acid (%)	1.99	LC-HR MS

**Table 4 ijms-26-08404-t004:** IC_50_ values of purwoceng extract antioxidants against DPPH and ABTS.

Sample	IC_50_ (mg/L)	
	DPPH	ABTS
Purwoceng extract	376.88	1269.77
Vitamin C	4.70	36.90

**Table 5 ijms-26-08404-t005:** BLASTp results of pro-oxidant enzymes from *Mus musculus* and their orthologs.

Enzyme	Organism	Accession	Query Cover (%)	E-Value	Per. Ident (%)
NOX5	*Cylindrospermum stagnale*	K9WT99	95	7 × 10^−49^	34.86
	*Mus musculus*	AAI72138.1			
iNOS	*Homo sapiens*	P35228	100	0	87.26
	*Mus musculus*	AAC52356.1			
COX2	*Homo sapiens*	P35354	100	0	88.22
	*Mus musculus*	NP_035328.2			

**Table 6 ijms-26-08404-t006:** Suppose the results between enzymes from *Mus musculus* and their orthologs.

Enzyme	PDB_ID	Organism	Superpose (Å)
NOX5	8CAO	*Cylindrospermum stagnale*	2.391
	6WXR	*Mus musculus*	
iNOS	3E7G	*Homo sapiens*	0.385
	4UX6	*Mus musculus*	
COX2	5F1A	*Homo sapiens*	0.39

## Data Availability

All relevant data supporting the findings of this study are contained within this article.

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
