# Peer review of "Phytochemicals from Purwoceng (Pimpinella pruatjan) and Their Potential in Chronic Disease Prevention: Focus on Kidney Health"

_ijms, 2025, doi:10.3390/ijms26178404_

Round 1
Reviewer 1 Report
Comments and Suggestions for Authors
This manuscript entitled "Phytochemicals from Purwoceng (Pimpinella pruatjan) and Their Potential in Chronic Disease Prevention: Focus on Kidney Health" is interesting and all the results are significant. However there are some comments which should be corrected or revised before accepted.
- The article provides a detailed introduction about the plant origin, chemical components, and potential pharmacological effects of Purwoceng, and points out that its potential in antioxidant and anti-inflammatory properties may help alleviate kidney damage. However, the description of the differences and innovations of this research compared to other related studies is not clear enough. The authors are advised toemphasize unique aspects, such as compound screening, or mechanism exploration.
- The experimental design was reasonable, involving a control group, a model group, and different treatment groups at doses of 20 mg/kg BW, 30 mg/kg BW, and 40 mg/kg BW. However, the basis for choosing each dose was not detailed. It is recommended to provide theoretical derivation of the dose gradient or cite relevant references.
- The histopathological results indicated that a 20 mg/kg BW dose significantly improved the glomerular damage induced by cisplatin, while higher doses (30 - 40 mg/kg BW) led to deterioration of renal function. This finding suggests the importance of dose adjustment, butthe reason for the increased nephrotoxicity in the high-dose group has not been elaborated.
- Table 1 contains a lot of information. It is recommended to present the information separately or add annotations to facilitate readers' quick understanding.
- Fig.3 needs to be supplemented with scales.
- The molecular docking diagrams of enzyme target molecules in Figure 4 and Figure 5 are complex. It is suggested to add brief explanations to help non-professional readers understand the key information.
- English must be edited before resubmission.
Author Response
Dear The Editor,
International Journal of Molecular Sciences.
Title: Phytochemicals from Purwoceng (Pimpinella pruatjan) and Their Potential in Chronic Disease Prevention: Focus on Kidney Health
Thank you so much for the positive comments from all reviewers, it makes our manuscript better. Here are some of our responds for the reviewer comments:
|
Reviewer Comments |
Author responds |
Revision in Line, section in manuscript |
|
Reviewer 1 |
||
|
The article provides a detailed introduction about the plant origin, chemical components, and potential pharmacological effects of Purwoceng, and points out that its potential in antioxidant and anti-inflammatory properties may help alleviate kidney damage. However, the description of the differences and innovations of this research compared to other related studies is not clear enough.
|
Most published reports on purwoceng (P. pruatjan) focus on cultivation, phytochemical composition, general antioxidant capacity, or reproductive effects not on kidney protection in a disease model. Our study is the first to evaluate purwoceng extract in a cisplatin-induced nephrotoxicity model, assessing renal function (urea, creatinine), kidney weight, and detailed glomerular tubular histopathology, with confirmation in silico prediction test. |
Line 43-54;101-108 |
|
The authors are advised to emphasize unique aspects, such as compound screening, or mechanism exploration. The experimental design was reasonable, involving a control group, a model group, and different treatment groups at doses of 20 mg/kg BW, 30 mg/kg BW, and 40 mg/kg BW. However, the basis for choosing each dose was not detailed. It is recommended to provide theoretical derivation of the dose gradient or cite relevant references.
|
Our a priori dose window was set below the sub-chronic toxicity threshold reported by Arjadi et al., where P. pruatjan root extract induced hepatotoxicity at 42 mg/kg BW. We therefore tested 20, 30, and 40 mg/kg BW/day as an escalating gradient that probes efficacy below the toxicity signal and brackets the upper bound just under the prior toxic dose. |
Line 506-509 |
|
The histopathological results indicated that a 20 mg/kg BW dose significantly improved the glomerular damage induced by cisplatin, while higher doses (30 - 40 mg/kg BW) led to deterioration of renal function. This finding suggests the importance of dose adjustment, but the reason for the increased nephrotoxicity in the high-dose group has not been elaborated.
|
Together, our findings support that 20 mg/kg sits within the therapeutic window where antioxidant/anti-inflammatory activity predominates, restoring glomerulpatency and attenuating tubular injury. Escalation to 30–40 mg/kg likely crosses into a pro-oxidant/accumulative exposure regime in proximal tubules, unmasking nephrotoxicity and negating benefit. This dose–response is coherent with both our histopathology/biochemistry and the known renal handling and redox behavior of the extract’s lipophilic phytochemicals |
Line 402-404 |
|
Table 1 contains a lot of information. It is recommended to present the information separately or add annotations to facilitate readers' quick understanding. |
Thank you for your recommendation. We have separated Table 1 into 2 tables |
120-121 |
|
Fig.3 needs to be supplemented with scales. |
We have added the scale into Fig.3 |
205 |
|
The molecular docking diagrams of enzyme target molecules in Figure 4 and Figure 5 are complex. It is suggested to add brief explanations to help non-professional readers understand the key information. |
We appreciate this suggestion and agree that docking diagrams can be challenging for non-specialist readers. In the revised manuscript, we have added concise explanatory sentences to the figure legends and Results section to highlight the essential information conveyed by Figures 4 and 5 (now Figure 5 and 6). |
293-308 |
|
English must be edited before resubmission |
we had use the English editor from MDPI |
|

Reviewer 2 Report
Comments and Suggestions for Authors
This paper focuses on the effect of Purwoceng extract on cisplatin-induced nephrotoxicity. It adopts a relatively systematic design combining in vivo and in vitro experiments with computer simulation, and finds that the 20mg/kg dose has a nephroprotective effect, with pinellic acid possibly being the key component, thus having certain reference value. However, there are problems such as insufficient basis for dose selection, lack of analysis of unknown compounds, inadequate explanation of the mechanism of high-dose toxicity, and missing details of some experiments. The details are as follows:
1.Abstract:
In "moderate doses of the Purwoceng extract complied with safety standards", "moderate doses" does not clearly correspond to specific doses (20mg/kg is an effective protective dose in the paper, while 30-40mg/kg shows toxicity), which is easy to cause confusion;
In addition, only "pinellic acid showed strong affinity for COX-2 and iNOS" is mentioned, but the causal relationship between the molecular docking results and the in vivo "glomerular structure improvement" is not linked.
Introduction:
The opening mentions that Purwoceng "increases ejaculation and penetration" ([1-3]), but fails to explain the connection between this traditional effect and "kidney health", which is irrelevant to the core research on "prevention of chronic kidney disease".
It does not clarify why the "cisplatin-induced nephrotoxicity" model was chosen and the uniqueness of Purwoceng compared with existing interventions (such as curcumin and vitamin E).
2.Results:
Table 1: In heavy metal detection, "<detection limit" does not mark the LOD values of all elements (only Cd mentions LOD=0.00011 mg/kg, while Pb and As are not specified). Moreover, "Malaysia Canada Singapore Thailand" in the "Standards permitted by" column is formatted in a confusing way, with inconsistent units.
The 8 "unknown compounds" do not have their possible categories inferred, and the association between the most abundant "juniperic acid" (7.28%) and biological activity is not explained.
Table 3: The DPPH/ABTS IC50 of Purwoceng extract is much higher than that of vitamin C (indicating weak antioxidant capacity), but the subsequent discussion emphasizes its "antioxidant activity", which is contradictory.
"5 rats died during cisplatin induction" does not specify the group and cause of death, which may affect the balance between groups.
Histopathological scoring: It is not stated whether the glomerular/tubular injury scoring was performed blindly (without knowing the sample grouping), which may introduce bias.
It is not specified whether COX-2 (5F1A) and iNOS (3E7G) are rat homologous protein structures, and sequence differences between human-derived enzymes and rat targets may affect the relevance of docking.
The clustering analysis of docking conformations (such as the proportion of conformations with RMSD < 2Å) and the standard deviation of binding energy are not reported, making it difficult to judge the stability of the results.
3.Discussion:
Only "lipophilic compounds accumulate in proximal tubules" is speculated, but it is not verified with the lipid solubility parameters of compounds in Table 2 (such as linoleic acid with LogP≈7.0).
Purwoceng has weak in vitro antioxidant capacity (Table 3), but the discussion emphasizes both "antioxidant and anti-inflammatory properties" without clarifying their priority.
There is no comparison with the activity of congeneric plants (such as Pimpinella anisum) or known nephroprotective agents, making it impossible to highlight the advantages of Purwoceng.
In addition, limitations such as "differences between animal models and humans", "influence of unknown compounds", and "unevaluated long-term toxicity" are not discussed.
4.Materials and Methods:
Missing details of experimental operations: "maceration for 24 hours at room temperature" does not specify details such as whether it is protected from light or stirred, which affects the reproducibility of the extract;
The administration sequence of Purwoceng and cisplatin (such as pretreatment before cisplatin vs. simultaneous administration) is not clarified, making it impossible to distinguish between "prevention" and "treatment" effects;
"Sample size based on Federer's formula" does not specify the specific calculation parameters.
5.Conclusion:
The expression is too absolute: "Purwoceng is thought to play a role in improving kidney function" does not limit the effective dose (only 20mg/kg) and ignores high-dose toxicity.
6.Other issues:
Charts can be further optimized; the first appearance of "CKD", "iNOS", etc. needs to be in full name; "ureum" is a spelling error.
Author Response
Bogor, 18th August, 2025
Dear The Editor and reviewer,
International Journal of Molecular Sciences.
Title: Phytochemicals from Purwoceng (Pimpinella pruatjan) and Their Potential in Chronic Disease Prevention: Focus on Kidney Health
Thank you so much for the positive comments from all reviewers, it makes our manuscript better. Here are some of our responds for the reviewer comments:
|
Reviewer 2 |
||
|
Abstract |
||
|
In "moderate doses of the Purwoceng extract complied with safety standards", "moderate doses" does not clearly correspond to specific doses (20mg/kg is an effective protective dose in the paper, while 30-40mg/kg shows toxicity), which is easy to cause confusion;
In addition, only "pinellic acid showed strong affinity for COX-2 and iNOS" is mentioned, but the causal relationship between the molecular docking results and the in vivo "glomerular structure improvement" is not linked. |
We thank the reviewer for pointing this out. We have removed the term “moderate doses”
Docking indicated that pinellic acid forms multiple stabilizing interactions within the catalytic sites of COX-2 and iNOS, consistent with potential dual inhibition (Figures 5–6). Given the established iNOS→COX-2 crosstalk via S-nitrosylation that amplifies prostanoid production, concurrent attenuation of these enzymes is expected to reduce nitro-oxidative and inflammatory stress within the glomerular microvasculature. We thank you for your suggestion, we have already connected the causal relationship between the molecular docking results and the in vivo "glomerular structure improvement" in the abstract section |
24
28-32 |
|
Introduction |
||
|
The opening mentions that Purwoceng "increases ejaculation and penetration" ([1-3]), but fails to explain the connection between this traditional effect and "kidney health", which is irrelevant to the core research on "prevention of chronic kidney disease"
|
However, these traditional claims are not endpoints of our study and are not used to infer renal benefit. To avoid any confusion with our core objective, kidney protection. We have reframed the opening to a single neutral clause about ethnomedicinal use, and explicitly connected the study’s rationale to kidney-relevant mechanisms (nitro-oxidative pathways involving iNOS/COX-2/NOX) and to our in vivo outcomes (improved glomerular structure at 20 mg/kg BW) and in silico findings (pinellic acid affinity to COX-2 and iNOS). These links are now highlighted and fully referenced in the Introduction |
43-54 |
|
It does not clarify why the "cisplatin-induced nephrotoxicity" model was chosen and the uniqueness of Purwoceng compared with existing interventions (such as curcumin and vitamin E). |
Mechanistic fit to our hypothesis. Cisplatin reliably produces acute kidney injury that is driven by oxidative stress and inflammation (ROS/RNS generation, NOX activation, COX-2/iNOS up-regulation). These are precisely the axes our LC-MS and docking suggested Purwoceng constituents (notably pinellic acid and oxidized unsaturated fatty acids) could modulate. Using this model, therefore, offers strong mechanistic coherence between the disease drivers and the predicted targets of the extract. |
60-75 |
|
Results |
||
|
Table 1: In heavy metal detection, "<detection limit" does not mark the LOD values of all elements (only Cd mentions LOD=0.00011 mg/kg, while Pb and As are not specified). Moreover, "Malaysia Canada Singapore Thailand" in the "Standards permitted by" column is formatted in a confusing way, with inconsistent units. |
We thank the reviewer for pointing this out. We have revised Table 1 |
Tabel 1, line 120 |
|
The 8 "unknown compounds" do not have their possible categories inferred, and the association between the most abundant "juniperic acid" (7.28%) and biological activity is not explained. |
We thank the reviewer for noting the unannotated LC‑MS peaks and the high relative abundance of juniperic acid. We have revised the manuscript about juniperic acid in the discussion section |
329-333 |
|
Table 3: The DPPH/ABTS IC50 of Purwoceng extract is much higher than that of vitamin C (indicating weak antioxidant capacity), but the subsequent discussion emphasizes its "antioxidant activity", which is contradictory. |
We thank the reviewer for pointing out this important nuance. We agree that the absolute radical-scavenging activity of Purwoceng extract (DPPH/ABTS IC₅₀ in the mg/mL range) is much weaker than vitamin C, a potent pure antioxidant standard. We have revised the manuscript to clarify this apparent contradiction |
374-377 |
|
"5 rats died during cisplatin induction" does not specify the group and cause of death, which may affect the balance between groups. |
We thank the reviewer for this valuable comment. We have revised the manuscript to provide a clear description of mortality. |
142-151 |
|
Histopathological scoring: It is not stated whether the glomerular/tubular injury scoring was performed blindly (without knowing the sample grouping), which may introduce bias. |
We thank the reviewer for this valuable comment. We have revised the manuscript to provide a clear description related avoiding bias |
357-359 |
|
It is not specified whether COX-2 (5F1A) and iNOS (3E7G) are rat homologous protein structures, and sequence differences between human-derived enzymes and rat targets may affect the relevance of docking.
|
Although the docking receptors (5F1A, 3E7G) are human, we show high human–rodent sequence identity (≈88% COX-2; ≈87% iNOS), near-identical 3D folds (RMSD ≈0.39 Å), and conserved catalytic/anchoring residues at the binding pockets; together with low redocking RMSDs, this supports the translational relevance of our docking to the rat targets used in vivo |
228-229 |
|
The clustering analysis of docking conformations (such as the proportion of conformations with RMSD < 2Å) and the standard deviation of binding energy are not reported, making it difficult to judge the stability of the results. |
We thank the reviewer for this valuable comment. We have revised the manuscript to add the clustering analysis of docking conformations. |
237-250 |
|
Discussion |
||
|
Only "lipophilic compounds accumulate in proximal tubules" is speculated, but it is not verified with the lipid solubility parameters of compounds in Table 2 (such as linoleic acid with LogP≈7.0).
|
We thank the reviewer for this valuable comment. We have added the Log P of each compound. |
Table S1 supplementary |
|
Purwoceng has weak in vitro antioxidant capacity (Table 3), but the discussion emphasizes both "antioxidant and anti-inflammatory properties" without clarifying their priority.
|
The modest DPPH scavenging activity of Purwoceng is offset by its targeted inhibition of pro-inflammatory mediators (COX-2/iNOS), which aligns with its nephroprotective effects observed in vivo. This dual mechanism, modest direct antioxidant action coupled with potent anti-inflammatory activity, highlights its therapeutic potential in chronic kidney disease. |
328-329;374 |
|
There is no comparison with the activity of congeneric plants (such as Pimpinella anisum) or known nephroprotective agents, making it impossible to highlight the advantages of Purwoceng.
|
We appreciate the reviewer's insightful comment regarding comparative efficacy. Our study demonstrates that P. pruatjan (Purwoceng) offers distinct advantages over both congeneric species and established nephroprotective agents |
334-342
|
|
In addition, limitations such as "differences between animal models and humans", "influence of unknown compounds", and "unevaluated long-term toxicity" are not discussed. |
We thank the reviewer for pointing this out. We have added the limitation |
410-412 |
|
Materials and Methods |
||
|
Missing details of experimental operations: "maceration for 24 hours at room temperature" does not specify details such as whether it is protected from light or stirred, which affects the reproducibility of the extract; |
We thank the reviewer for pointing this out. We have added the missing detail of experimental operation in case of maceration |
429-430 |
|
The administration sequence of Purwoceng and cisplatin (such as pretreatment before cisplatin vs. simultaneous administration) is not clarified, making it impossible to distinguish between "prevention" and "treatment" effects; |
We thank the reviewer for pointing this out. We have already added the explanation |
506-509 |
|
"Sample size based on Federer's formula" does not specify the specific calculation parameters. |
We sincerely appreciate the reviewer's request for clarification on our sample size determination. We adopted Federer’s method because: - It is widely used in animal experiments where pilot data for variance is limited. - It ensures sufficient replication for ANOVA validity while adhering to ethical principles of using the minimum number of animals required. |
|
|
Conclusion |
||
|
The expression is too absolute: "Purwoceng is thought to play a role in improving kidney function" does not limit the effective dose (only 20mg/kg) and ignores high-dose toxicity |
We thank the reviewer for pointing this out. We have revised the explanation |
542-544 |
Improvements or parts of the manuscript that have been updated are highlighted in red.
Thank you very much,
Sincerely
Irmanida Batubara

Reviewer 3 Report
Comments and Suggestions for Authors
Dear authors, your investigation into the nephroprotective properties of Purwoceng (Pimpinella pruatjan) is potentially interesting, however, I think it has some drawbacks:
While in silico results suggest COX-2 and iNOS interactions, the in vivo mechanisms (e.g., specific antioxidant pathways, gene/protein expression) are not validated. There is no measurement of oxidative stress markers (e.g., MDA, SOD, GPx) in the kidney tissue to support the antioxidant hypothesis.
The 20 mg/kg BW dose is reported as nephroprotective, while higher doses cause harm. However, the explanation for this non-linear response is speculative, attributing it to lipophilic compound accumulation without any pharmacokinetic or tissue distribution data. No toxicological studies or histopathological scoring systems were used to validate the supposed nephrotoxicity at higher doses.
My suggestions for improvement are:
Use a blinded histopathologist for kidney scoring. Test kidney antioxidant enzymes (SOD, catalase). Include ELISA or qPCR for COX-2, iNOS expression.
Author Response
Bogor, 18th August, 2025
Dear The Editor and reviewer,
International Journal of Molecular Sciences.
Title: Phytochemicals from Purwoceng (Pimpinella pruatjan) and Their Potential in Chronic Disease Prevention: Focus on Kidney Health
Thank you so much for the positive comments from all reviewers, it makes our manuscript better. Here are some of our responds for the reviewer comments:
|
Reviewer 3 |
||
|
While in silico results suggest COX-2 and iNOS interactions, the in vivo mechanisms (e.g., specific antioxidant pathways, gene/protein expression) are not validated. There is no measurement of oxidative stress markers (e.g., MDA, SOD, GPx) in the kidney tissue to support the antioxidant hypothesis.
|
We thank the reviewer for this important point. Our study did not quantify renal oxidative-stress markers (e.g., MDA, SOD, GPx) nor gene/protein expression of COX-2, iNOS, or NOX in kidney tissue. As now stated in the revised manuscript, our in vivo endpoints were renal function (serum urea/creatinine), kidney weight, and blinded PAS histopathology scoring, and our mechanistic discussion is therefore putative, guided by docking and the known biology of cisplatin nephrotoxicity |
not conducted in this study
Table S1 supplementary
|
|
The 20 mg/kg BW dose is reported as nephroprotective, while higher doses cause harm. However, the explanation for this non-linear response is speculative, attributing it to lipophilic compound accumulation without any pharmacokinetic or tissue distribution data. No toxicological studies or histopathological scoring systems were used to validate the supposed nephrotoxicity at higher doses. |
We thank the reviewer for this important point. We agree that the manuscript’s mechanistic explanation for why 20 mg/kg BW was nephroprotective while 30–40 mg/kg BW appeared harmful is currently speculative and that definitive attribution (e.g., lipophilic compound accumulation) requires dedicated pharmacokinetic (PK), tissue-distribution and toxicology data. We have therefore tempered the Discussion and added a clear limitations paragraph |
364-373; 393-403; 410-418 |
|
My suggestions for improvement are:
Use a blinded histopathologist for kidney scoring. Test kidney antioxidant enzymes (SOD, catalase). Include ELISA or qPCR for COX-2, iNOS expression. |
We thank the reviewer for this important point and suggestion |
522-524 |
Improvement or Part of manuscript that has been updated are highlighted in red.
Thank you very much,
Sincerely
Irmanida Batubara

Round 2
Reviewer 1 Report
Comments and Suggestions for Authors
Please check the caption of Picture 3, and change "g" to "e" in the caption.
Author Response
Comment from Reviewer: Please check the caption of Picture 3, and change "g" to "e" in the caption.
Author responds: We had deleted s from groups into group in the caption of Figure 3, and we had changed the g to e in Figure 3, line 208.
Thank you so much for your corrections to improve the manuscript.
Reviewer 2 Report
Comments and Suggestions for Authors
The author has already solved all my problems.
Author Response
Comment 1: The author has already solved all my problems.
Respond: Thank you very much for your valuable suggestion.